# Sleep-dependent memory consolidation in infants protects new episodic memories from existing semantic memories

Manuela Friedrich [1,2✉], Matthias Mölle [3], Angela D. Friederici[2] & Jan Born [4]

Any experienced event may be encoded and retained in detail as part of our episodic memory, and may also refer and contribute to our generalized knowledge stored in semantic memory. The beginnings of this declarative memory formation are only poorly understood. Even less is known about the interrelation between episodic and semantic memory during the earliest developmental stages. Here, we show that the formation of episodic memories in 14- to 17-month-old infants depends on sleep, subsequent to exposure to novel events. Infant brain responses reveal that, after sleep-dependent consolidation, the newly stored events are not processed semantically, although appropriate lexical-semantic memories are present and accessible by similar events that were not experienced before the nap. We propose that temporarily disabled semantic processing protects precise episodic memories from interference with generalized semantic memories. Selectively restricted semantic access could also trigger semantic refinement, and thus, might even improve semantic memory.

[1] Department of Psychology, Humboldt-University of Berlin, Rudower Chaussee 18, D-12489 Berlin, Germany. [2] Department of Neuropsychology, Max Planck Institute for Human Cognitive and Brain Sciences, Stephanstraße 1a, D-04103 Leipzig, Germany. [3] Center of Brain, Behavior and Metabolism (CBBM), University of Lübeck, Marie-Curie-Straße, D-23562 Lübeck, Germany. [4] Institute of Medical Psychology and Behavioral Neurobiology and Center for Integrative Neuroscience, University of Tübingen, Otfried-Müller-Str. 25, D-72076 Tübingen, Germany. ✉email: friedri@cbs.mpg.de

Humans are able to mentally re-experience past events in a very detailed manner, and likewise they are able to extract the gist of an experience and generalize it to novel circumstances. The two types of memory that are specialized for these competencies, episodic and semantic memory, are part of the hippocampus-dependent declarative memory system. Even though the full maturation of their relevant brain structures extends into adolescence[1,2], main abilities necessary to form basic episodic and semantic memories are developed in the first half of the second year of life[3–8].

In adults, episodic and semantic memories complement each other, but they also interact. Episodic memories represent a main source for semantic generalizations and, hence, are relevant for the formation and modification of semantic memories[9–11]. Semantic memories, on the other hand, may affect the encoding of episodic experience. Lexicalized semantic memories are moreover necessary for the verbal retrieval of episodes. Importantly, the potential access to episodic and semantic memories overlaps. For example, a certain word occurring in close temporal relation with a specific object may be encoded and retained as a unique event in episodic memory, while, at the same time, it may represent the labelling of a more general category stored in semantic memory. The conditions necessary for the formation of earliest episodic and semantic memories are largely unknown, and the interrelation of these memories in infants has not yet been investigated.

The contribution of an experience to a certain kind of memory depends on both the encoding of relevant information and the consolidation of immediately formed memories. In the mature brain, memory consolidation requires a process, during which transient memories are replayed and transformed into more stable representations. A large body of research supports the view that episodic and semantic memories are replayed during Non-REM sleep (NonREM for non-rapid eye movement)[12–20]. Sleep spindles, brief oscillatory neural events with waxing and waning amplitude at a frequency of 11–15 Hz, are involved in the hippocampal-neocortical dialogue during memory replay and presumably induce the synaptic changes that underlie sleep-dependent memory plasticity in the neocortex[21–27].

NonREM sleep with a substantial amount of central-parietal fast sleep spindles in the 13–15 Hz frequency range supports semantic generalization of memories even in infants[28–30]. The amount of central-parietal fast spindle activity, in turn, is affected by an infant's processing of novel experience immediately before a nap. The less generalized knowledge is available for novel experience—either in semantic long-term memory or transiently formed during encoding—the higher the central-parietal spindle activity in a subsequent nap. In turn, the higher the spindle activity during the nap, the stronger the semantic generalization of the novel experience in the memory test on a next day[30].

Together, these findings highlight the reciprocal relations between infant sleep spindle activity and early semantic memory. To date, however, nothing is known about the impact of sleep on the consolidation of episodic memories in early infancy.

In the present study we explore whether sleep in 14- to 17-month-old infants supports the consolidation of individual object–word pairings as specific episodic-like memories, and whether sleep spindles are involved in this consolidation process. We also aim to assess the interaction of related episodic and semantic memories during earliest developmental stages. We use a study design, in which 14–17-month-old infants are exposed to a set of object–word pairs, for which lexical-semantic memories are expected to be present at that age. More specifically, objects that represent exemplars of first acquired basic-level categories are presented together with their labelling words (Fig. 1). In the retention period subsequent to encoding, infants of one group nap while infants of the other group stay awake. After the retention period, memory is tested by exposing infants to old objects that have been experienced before the nap and new objects that have not been experienced previously. Each object is presented twice: once paired with the correct word and once paired with an incorrect word that match one of the other categories.

Importantly, objects are always presented first, such that an object serves as a context, in which the word is temporally embedded. This design allows us to capture expectations generated by context-dependently activated memory representations. Memory is assessed by analysing event-related potentials (ERPs) time-locked to words. The presence of semantic long-term memory is inferred from the N400 component in the ERP[31,32]. The N400 is known to be reduced in response to stimuli that are embedded in semantically related contexts and to be pronounced for stimuli that occur in semantically unrelated contexts. Based on previous infant studies[6,28,33–37], an N400 semantic context effect is expected to manifest as ERP difference between incorrectly and correctly paired words, if appropriate lexical-semantic memories are available and activated by the object context. Crucially, old and new objects are exemplars of the same categories and words are the same in all conditions. Thus, any N400 effect would indicate the existence of lexical-semantic memory for these same word stimuli. Differences in the N400 between old and new objects are attributed to the impact of recent episodic experience on the semantic processing of that experience. In order to assess episodic memory, we identify the specific processing distinctions between words occurring in their old object contexts and the same words appearing in other old or in new object contexts.

As a result, we provide clear evidence that, even in 1-year-olds, sleep is crucial for the consolidation of episodic memories. Infants who stay awake after encoding, do not retain detailed memories for the individual pairings of objects and words. In contrast,

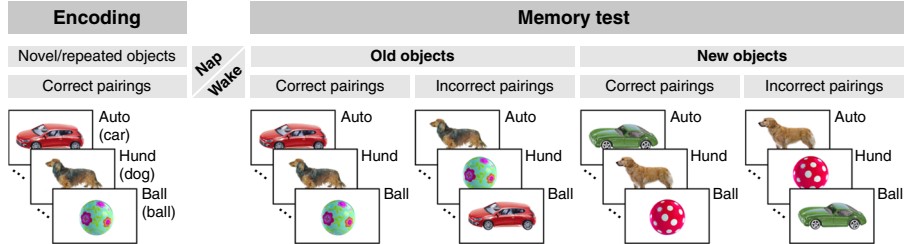

**Fig. 1 The experimental design.** In the encoding session, infants were exposed to known words presented in semantically correct object contexts. Each initially novel object–word pair was repeated once within about a minute. In the retention period, half of the infants napped while the other half stayed awake. In the memory test after the retention period, infants were re-exposed to the same words. The words were now presented both in the same episodic context as during encoding, i.e., with old semantically correct objects, and in three different contexts: with old but semantically incorrect objects, with new semantically correct objects, and with new semantically incorrect objects.

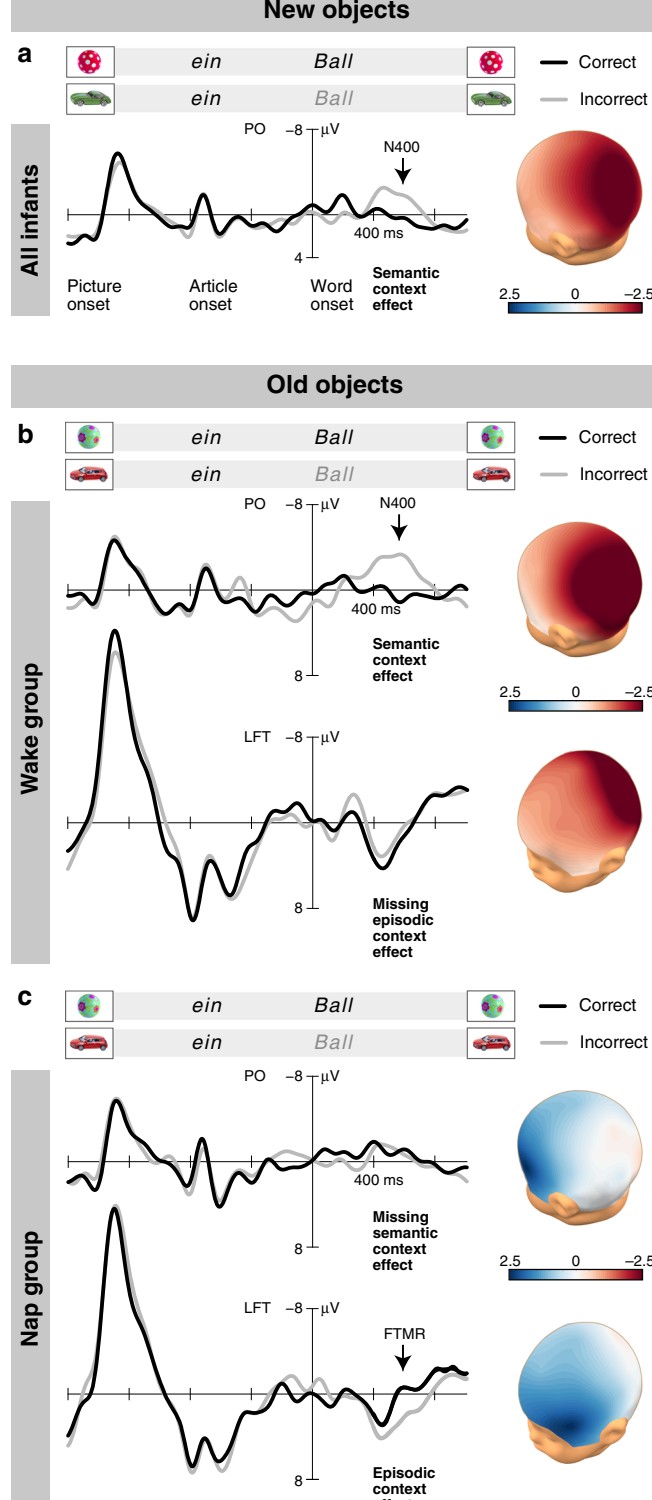

**Fig. 2 The N400 semantic memory effect and the FTMR episodic memory effect.** ERP responses to correct (black lines) and incorrect (grey lines) object–word pairs in the whole trial interval, time-locked to word onset. Negativity is plotted upward. The parieto-occipital (PO) region includes mid-parieto-occipital, left parietal, and right parietal ROIs and was calculated as the mean of the ERP amplitudes at the electrode sites P3, PZ, P4, CP5, CP6, P7, P8, O1, and O2. The left fronto-temporal ROI (LFT) involves F7 and T7. Voltage maps (colour scale in µV) represent the spatial distributions of the ERP differences between incorrectly paired and correctly paired words in the temporal range between 400 and 800 ms post word onset. **a** N400 semantic context effect to pairings with new objects across wake and nap groups indicating semantic memory ($t_{59} = -5.123$, $P < 0.0001$, $d = -0.661$, one-sample $t$-test). **b** N400 semantic context effect to pairings with old objects in the wake group indicating semantic processing of the old events ($t_{29} = -3.632$, $P = 0.001$, $d = -0.663$, one-sample $t$-test) and missing episodic context effect ($t_{29} = -0.863$, $P = 0.395$, one-sample $t$-test). **c** Episodic context effect over the left fronto-temporal region (FTMR) to pairings with old objects in the nap group indicating episodic memory ($t_{29} = 3.526$, $P = 0.001$, $d = 0.644$, one-sample $t$-test), and missing N400 semantic context effect ($t_{29} = 0.238$, $P = 0.814$, one-sample $t$-test). The words "*ein*" and "*Ball*" are italicised to represent the speech presented in the experiment. Context effects are indicated in bold. Source data are provided as a Source Data file.

## Results

**Word processing is affected by sleep and context conditions.** Repeated measures ANOVA of the ERPs to words presented during the memory test revealed processing differences between infant brain responses to incorrectly and correctly paired words (Semantic Congruity $F_{1,58} = 7.625$, $P = 0.008$, $\eta_p^2 = 0.116$, Semantic Congruity × Region $F_{1,58} = 16.143$, $P = 0.0002$, $\eta_p^2 = 0.218$, Semantic Congruity × Laterality $F_{2,116} = 5.655$, $P = 0.007$, $\eta_p^2 = 0.089$), which to the greatest part reflects the expected N400 semantic context effect. These processing differences were modulated by post-encoding sleep (Semantic Congruity × Group $F_{1,58} = 6.166$, $P = .016$, $\eta_p^2 = 0.096$) and were also dependent on the specific episodic context in which the words were presented during the learning session (Old/New × Semantic Congruity × Group $F_{1,58} = 5.175$, $P = 0.027$, $\eta_p^2 = 0.082$). Subsequent analyses were performed to disentangle the contributions of these two factors.

**Semantic memories were present in infants of both groups.** For pairings with new objects, which the infants had not experienced in the encoding session, a pronounced N400 semantic context effect was observed over parieto-occipital regions (mid-parieto-occipital: $t_{59} = -5.452$, $P < 0.0001$, $d = -0.681$, left parietal: $t_{59} = -3.510$, $P = 0.001$, $d = -0.576$, right parietal: $t_{59} = -2.796$, $P = 0.007$, $d = -0.528$, paired $t$-tests; Supplementary Fig. 1a). The overall effect (Fig. 2a) was present in both groups (wake: $t_{29} = -3.992$, $P = 0.001$, $d = -0.729$, nap: $t_{29} = -3.239$, $P = 0.003$, $d = -0.591$, one-sample $t$-tests) and did not differ in amplitude between them ($t_{58} = -1.172$, $P = 0.246$, $t$-test for independent samples). Within the investigated age range, it was not correlated with the infants' age ($r = -0.110$, $P = 0.401$, Pearson's $r$). It was also not correlated with the duration of the retention period, neither in the nap group ($r = 0.123$, $P = 0.519$) nor in the wake group ($r = 0.082$, $P = 0.665$). The occurrence of this N400 context effect clearly attests the existence of appropriate lexical-semantic memories of the stimulus material and the similar accessibility of these memories in the infants of the wake and nap groups.

infants who nap after encoding, retain this information. Their memory for the episodic context of a word is the better, the higher the amplitude of their frontal fast sleep spindles during the post-encoding nap. Differences in the N400 between words paired with old objects and words paired with new objects reveal that, after the nap, the new memory for an episodic experience interferes with the semantic processing of that experience, a mechanism that might enable both precise retrieval and semantic refinement.

**Semantic processing of old events differed between groups**. In the wake group, an N400 semantic context effect was also present for word pairings with old objects, which the infants had already experienced during the encoding session ($t_{29} = -3.632$, $P = .001$, $d = -0.663$, one-sample $t$-test; Fig. 2b, Supplementary Fig. 1b). In this group, the N400 effect for words occurring in old object contexts did not differ from the N400 effect for words in new object contexts ($t_{29} = 0.846$, $P = 0.404$, paired $t$-test). Thus, in infants who had stayed awake during the retention period, semantic word processing was not affected by the episodic context, in which a word had been presented in the encoding session.

In contrast, in infants who had napped during the retention period, the N400 semantic context effect was missing for words paired with old objects ($t_{29} = 0.238$, $P = 0.814$, one-sample $t$-test; Fig. 2c, Supplementary Fig. 1c). The deviating brain response in these infants (wake–nap group difference: $t_{58} = 3.285$, $P = 0.002$, $d = 0.785$, $t$-test for independent samples) suggests that sleep-related processes after the encoding of an object–word episode resulted in selectively restricted access to semantic memory when the word was re-experienced in the same episodic context subsequent to the nap.

**Episodic memories were formed in infants of the nap group**. Instead of the N400 semantic memory effect, but at about the same latency as the parietal N400 in the wake group, the nap group showed a distinct memory effect with inversed polarity and left fronto-temporal distribution for pairings with old objects ($t_{29} = 3.526$, $P = 0.001$, $d = 0.644$, one-sample $t$-test; Fig. 2c, Supplementary Fig. 1c). This fronto-temporal memory response (FTMR) did not occur for pairings with new objects ($t_{29} = -1.061$, $P = 0.298$, one-sample $t$-test). It was also not present in the wake group ($t_{29} = -0.863$, $P = 0.395$, one-sample $t$-test; wake–nap group difference: $t_{58} = -2.911$, $P = 0.005$, $d = -0.708$, $t$-test for independent samples; Fig. 2b). The FTMR of the nap group was neither correlated with the infants' age ($r = -0.100$, $P = 0.601$), nor with the duration of the retention period ($r = 0.098$, $P = 0.605$), total sleep time ($r = -0.182$, $P = 0.337$), or the time spent in individual sleep stages (Table 1; $|r| = 0.060$–$0.204$, $P = 0.280$–$0.753$). The effect resulted mainly from a more negative response to words paired correctly with old objects, which, after the nap, re-occurred in the same object contexts as during encoding (correct old vs. correct new: $t_{29} = -3.211$, $P = 0.003$, $d = -0.586$, paired $t$-test; Fig. 3b). Thus, the FTMR provides evidence of memory for the specific episodic context in which a word was experienced before the nap.

**Episodic memory did not linearly depend on object memory**. In order to explore how this memory for the episodic pairing of objects and words is related to the recognition memory of objects, we analysed the brain responses to the first presentation of an individual object in the memory test. An early-latency old/new memory effect over the occipital region (Old/New × Region $F_{3,174} = 8.338$, $P = 0.001$, $\eta_p^2 = 0.126$, repeated measures ANOVA; occipital $t_{59} = -3.505$, $P = 0.001$, $d = -0.452$, paired $t$-test; Supplementary Fig. 2b) indicated visual memory for the specific objects in infants of both groups (wake: $t_{29} = -2.489$, $P = 0.019$, $d = -0.454$, nap: $t_{29} = -2.678$, $P = 0.012$, $d = -0.489$, one-sample $t$-tests; group comparison: $t_{58} = -0.817$, $P = 0.417$, $t$-test for independent samples). A second old/new memory effect over frontal and central regions at a latency of 600–800 ms, which may reflect higher-level recognition memory, was present only in the nap group (Old/New × Group $F_{1,58} = 10.276$, $P = 0.002$, $\eta_p^2 = 0.151$, Old/New × Region $F_{3,174} = 4.294$, $P = 0.014$, $\eta_p^2 = 0.069$, repeated measures ANOVA, nap frontal: $t_{29} =$

Table 1 Sleep characteristics of the post-encoding nap.

|  | Mean | SD |
|---|---|---|
| Stage 1 sleep | 11.55 | 7.76 |
| Stage 2 sleep | 25.81 | 12.40 |
| Slow wave sleep | 19.67 | 10.70 |
| TST | 57.02 | 21.86 |
| Spindle RMS frontal | 11.27 | 3.08 |
| Spindle RMS central-parietal | 9.19 | 2.11 |

Stage 1 sleep, stage 2 sleep, slow wave sleep, and total sleep time (TST) in minutes, and spindle RMS amplitude in µV (frontal: mean of F3 and F4, central-parietal: mean of C3, C4, P3, and P4).

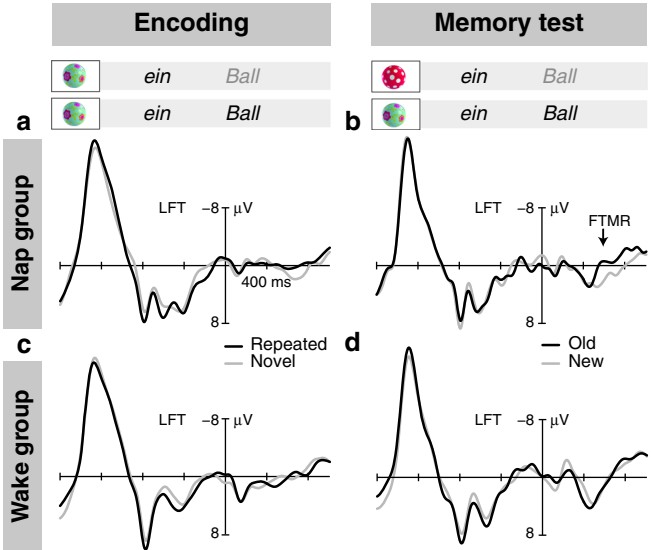

**Fig. 3 Immediate and later memory for the episodic pairings of objects and words. a** ERP responses of the nap group at encoding. Non-significant difference between the ERPs to words presented the second time in the same semantically correct object context (black lines) and the same words presented the first time in this specific object context (grey lines). **b** ERP responses of the nap group in the memory test. Old/new FTMR ($t_{29} = -3.211$, $P = 0.003$, $d = -0.586$, paired $t$-test) for words presented in old semantically correct object context (black lines) compared with the same words presented in a new semantically correct object context (grey lines). **c** ERP responses of the wake group at encoding. No significant difference over the LFT region. **d** ERP responses of the wake group in the memory test. Missing old/new FTMR ($t_{29} = -0.136$, $P = 0.893$, paired $t$-test) in the memory test of the wake group. Source data are provided as a Source Data file.

$-3.368$, $P = 0.002$, $d = -0.615$, nap central: $t_{29} = -3.733$, $P = 0.001$, $d = -0.682$, wake frontal: $t_{29} = 1.382$, $P = 0.178$, wake central: $t_{29} = 1.299$, $P = 0.204$, one-sample $t$-tests; group difference for the overall frontal-central effect: $t_{58} = 3.506$, $P = 0.001$, $d = 0.829$, $t$-test for independent samples; Supplementary Fig. 2b–d). This additional effect in the nap group was driven by a difference in the processing of old objects rather than new objects (group difference for old objects; $t_{58} = 2.779$, $P = 0.007$, $d = 0.680$, group difference for new objects: $t_{58} = -0.626$, $P = 0.534$, $t$-tests for independent samples). None of the old/new effects in response to the pictured objects were correlated with the FTMR episodic context effect in response to words (early occipital: $r = 0.243$, $P = 0.196$, late frontal-central: $r = 0.025$, $P = 0.894$), which suggests that memory for the episodic binding

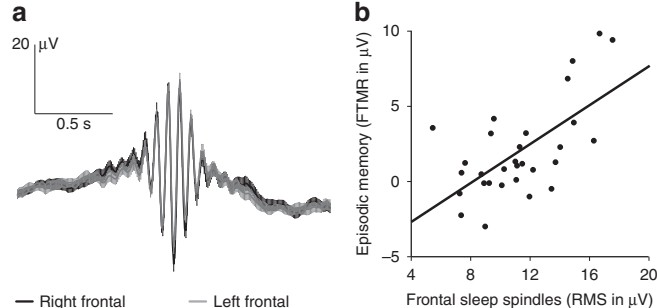

**Fig. 4 Frontal fast sleep spindles and memory for the episodic context. a** Fast sleep spindles during NonREM sleep of the post-encoding nap recorded over the left frontal (F3) and right frontal (F4) regions (mean ± SEM). **b** Correlation between the RMS amplitude of frontal fast sleep spindles (mean of F3 and F4) during the nap and the amplitude of the left FTMR for words presented in old object contexts in the memory test after the nap ($r = .636$, $P = 0.0002$, Pearson's $r$). Source data are provided as a Source Data file.

between an object and a word does not linearly depend on recognition memory for the individual object.

**Encoding did not directly affect the memory after the nap.** In a next step we explored how the strength of encoding affected infant memories. The repetition of individual objects during the encoding session was reflected in a reduction of the negativity in the 600–800 ms time window of the ERP (Repetition $F_{1,58} = 12.229$, $P = 0.001$, $\eta_p^2 = 0.174$, repeated measures ANOVA; Supplementary Fig. 2a). This effect was present in both groups (wake: $t_{29} = 2.326$, $P = 0.027$, $d = 0.425$, nap: $t_{29} = 3.185$, $P = 0.003$, $d = 0.581$ for the frontal-central difference, one-sample $t$-tests; Supplementary Fig. 2d) and did not differ between them ($t_{58} = 0.126$, $P = 0.900$, $t$-test for independent samples). The object repetition effect during encoding was not correlated with the frontal-central old/new object memory effect of the nap group, which had the same latency, but an inversed polarity ($r = -0.192$, $P = 0.310$). The corresponding (non-significant) old/new ERP difference in the memory test of the wake group, however, was correlated with the encoding repetition effect of this group ($r = 0.434$, $P = 0.016$; Supplementary Fig. 2c). This pattern suggests that the memory for individual objects formed immediately during the encoding session was, to some extent, retained in the infants of the wake group but underwent substantial modification during the retention phase of the nap group.

In contrast to object repetition, the episodic re-occurrence of words in the same object context did not induce a significant ERP effect in the encoding session ($P > 0.1$ for all comparisons involving Repetition in the repeated measures ANOVA; Fig. 3a and c). The responses over the left fronto-temporal region, in particular, did not differ between nap and wake groups ($t_{58} = 1.437$, $P = 0.156$ for the repeated–novel difference, $t$-test for independent samples), although in the nap group, repeated pairings tended to elicit a somewhat more negative response than novel pairings ($t_{29} = -1.719$, $P = 0.096$, paired $t$-test). We did not find a correlation between this non-significant tendency towards an FTMR-like response during encoding and the strength of the old/new FTMR in the memory test after the nap ($r = -0.036$, $P = 0.851$). This speaks against a simple direct relation between the immediate formation of an episodic memory representation and its subsequent strength. Rather, the intervening nap appears to be the crucial factor that strengthened the episodic binding between objects and words.

**Frontal sleep spindles are linked to episodic memories.** Not only was the FTMR to the recent episodic context exclusively present in the memory test of the nap group, it was also

correlated with the infant's spindle activity during NonREM sleep of the post-encoding nap. In particular, it was correlated with the root mean square (RMS) amplitude of fast sleep spindles over left and right frontal brain regions (left: $r = 0.618$, $P = 0.0003$, right: $r = 0.611$, $P = 0.0003$, Pearson's $r$; Fig. 4). In contrast, the old/new recognition effect for the individual objects was not related to spindle activity ($r = 0.127 – 0.257$, $P = 0.170 – 0.505$). This provides evidence for the view that recognition memory for a specific object, and episodic memory for the pairing of a specific object with a word, are supported by distinct consolidation mechanisms during infant sleep.

Finally, the correlation of spindle activity over central-parietal brain regions with the N400 effect for words paired with new objects, which had been observed in recent infant studies[28–30], failed to meet statistical significance after correcting for multiple testing (highest correlation with spindle RMS at P4: $r = -0.437$, $P = 0.016$, adjusted $\alpha = 0.008$; overall correlation with mean spindle RMS amplitude over the left and right central and parietal regions: $r = -0.174$, $P = 0.358$). This weak relation suggests that infants did not build substantial new semantic memories during their nap. Together with the presence of an N400 effect in the wake group, these results imply that the semantic memory effect in response to new objects was mainly based on the infants' pre-existing lexical-semantic knowledge.

## Discussion

Detailed retention of individual experience and generalization of similar experiences to novel events are human core abilities based on highly specified and complementary memory systems. Here we show that these abilities are both present in 14- to 17-month-old infants. A mid-latency fronto-temporal component in the ERP was found to indicate infant memory for the specific episodic context in which a word had occurred during the encoding session. This neural correlate of infant episodic memory, which we termed FTMR, was clearly distinct from the neural correlate of semantic memory, the N400 response dominating over the parietal cortex. It also differs in its spatio-temporal characteristics from the more posterior late negativity in the infant ERP, which has been reported in previous studies with younger infants and attributed to early developing perceptual-associative memories[29,38]. Instead, although occurring somewhat later, the FTMR shares spatial characteristics with the N200–500, an infant ERP response to words, which has been observed in several studies with uni- and bimodal stimulus presentations and can occur both in the absence and presence of an N400 response[33,34,38–44]. The N200–500 reflects a non-semantic word processing stage that is affected by stimulus repetition, familiarity, and expectation. Most likely, it is linked to

an acoustic-phonological stage of processing, such as the activation of acoustic-phonological features or their binding to a coherent word. If the FTMR represents an N200–500 response, this would imply that newly formed episodic memory representations for the specific pairings of objects and words facilitate acoustic-phonological word processing, when a word is re-experienced in its same episodic object context. A word in the context of an old object would then be processed as more familiar than the same word in the context of a new object. Whether the FTMR indeed reflects such a modality-specific effect of episodic memory on word processing, i.e., a kind of perceptual priming, or rather a modality-independent effect, such as the facilitated integration of the correct item into the current episode, which would represent a kind of "episodic priming", remains to be solved by future studies.

A similar pattern of ERP effects as in the present study has been found in the nap group of a previous learning study. In this study, 9- to 16-month-old infants were presented with pairings of objects and words, for which they had no prior lexical-semantic knowledge[28]. After the nap, pairings with novel objects in a general (category) learning condition elicited an N400 effect. However, pairings with previously observed objects in a specific learning condition led to an N200–500 effect, without a concurrent N400. The similarity of the N200–500 with the FTMR observed here suggests that, as in the present study, episodic memories were built in the specific learning condition of that previous study. In particular, the lack of object variability in the specific learning condition could have triggered the formation of episodic instead of semantic memories, while in the general learning condition with high object variability, new semantic memories were built. Since object variability in the present study was quite similar to that of the general learning condition in the previous study, here, the pre-existing lexical-semantic memory appears to be the crucial factor that led to the consolidation of episodic memories. From the data available so far, it remains open, whether the consolidation of a new semantic memory precludes the consolidation of a related episodic memory or whether these memories could even be built in parallel.

In the present study, evidence of episodic memory was only found in infants who had napped after initial exposure. Thus, even though infants of the wake group retained some visual information of the specific objects, as indicated by their occipital old/new object memory effect, sleep was crucial to retain the episodic binding of this information. Notably, episodic memory as reflected in the FTMR was related to sleep spindle activity during the nap, a finding that is consistent with several studies reporting an effect of sleep spindles on episodic memory consolidation in adults[45–47]. It also fits with previous findings on sleep-dependent memory consolidation in infants[28–30]. While, however, the formation of generalized semantic memories is linked to infant central-parietal spindle activity, here, the consolidation of episodic memories was associated with enhanced sleep spindles over frontal brain regions. Thus, we provide the first evidence that similar mechanisms of spindle-dependent cortical plasticity, but regionally distinct neural circuits during sleep, support the consolidation of either episodic or semantic memories.

A main novel finding of the present study was the interfering effect of infant episodic memories on the subsequent semantic processing of episodic events. In infants of the wake group, who had not retained recent episodic experience, the expected N400 effect indicated semantic processing of the object–word pairs, independent of whether the pairing involved an old or a new object. In contrast, in the nap group, in which the FTMR revealed the consolidation of new episodic memories, the N400 semantic context effect was missing, when old object–word pairs were re-experienced after the nap. However, the N400 effect to words in new object contexts provides evidence for the presence and accessibility of appropriate lexical-semantic memories in the infants of this group. Since words were the same in both conditions, and the N400 to new objects did not differ between wake and nap groups, the missing semantic context effect for words in old object contexts in the napping infants could not be caused by weaker or missing lexical-semantic memories. Why then did the words in the old object contexts not elicit the same semantic N400 effect as in the new object contexts?

Missing semantic context priming, despite appropriate lexical-semantic memories, may result from insufficient semantic processing of either the primes or the targets. In adults, semantic processing of words occurs passively through an automatic spread of activation, and the N400 word priming effect is eliminated only when the primes are not attended[32]. Accordingly, the objects, serving as the primes in the present study, may be responsible for the missing N400 effect. In the infants who had episodic memories available, the recognition of an old object might have shifted attention towards the recent episode, thereby omitting semantic processing of the object. Rather than pure omission, the focus on the recognized details of an old object might have dissociated the object from the generalized semantic representation, such that the assignment to this representation failed and semantic priming of the related word was prevented. Even if this unsuccessful memory access induced the formation of a more specific semantic representation, the immediately formed representation would not yet have been linked to a word. Thus, semantic word priming would not have been possible. It is also conceivable that old objects accessed existing semantic memories and triggered semantic word priming, but that priming was ineffective, i.e., the words as the targets may be responsible for the missing effect. In this scenario, semantic processing of the words was initiated, but episodic memories were activated in parallel and competed with semantic memory activation. As a result, semantic processing of the words was inhibited, such that semantic priming had no effect on word processing. From the present data it is not clear whether new episodic memories had already captured attention before semantic priming was initiated, or whether they inhibited semantic processing after semantic pre-activation. Irrespective of the underlying mechanisms, our findings strongly suggest that newly formed episodic memories in the nap group interfered with semantic processing and, at least temporarily, prevented access to existing semantic representations, when words were perceived in their recent episodic context.

At first glance, restricted access to semantic memory does not seem to be favourable. In a changing environment, however, the most recent experience may be particularly relevant for behavioural adaptation in the near future. Thus, fast direct access to recent information without the involvement of older memories might be advantageous. The unavailability of semantic memories may enable infants to re-experience recent events without any distortion by existing general knowledge and, thus, may protect precise episodic memories from interference with generalized semantic memories. Moreover, during earliest developmental stages, semantic long-term memory comprises extremely broad categories that act as strong attractors for a variety of objects and events and causes the well-known overgeneralizations in the comprehension and production of early words[48,49]. The temporary selective restriction of access to existing representations might provide an opportunity to overcome strong attractors by enabling the formation of more specific representations. This potential mechanism would ultimately boost the development of infant semantic memory.

Selective exclusion of assignment to existing memories can, in fact, be observed in the context of early lexical-semantic

specification. When young children have learned a new word as a label for a new sub-category of a known basic-level category, many of them subsequently avoid selecting objects of the sub-category as an instance of the basic-level category. For example, after learning the word 'zav' as a name for tulips, about half of the children no longer identified tulips as flowers[50]. This strong behavioural tendency to treat hierarchically related categories as mutually exclusive sub-categories, observed in somewhat older children, bears some similarity with the apparent mutually exclusive access to episodic and semantic memories observed in the present study. The precise relationships between the underlying memory processes and respective behavioural and ERP indicators still need to be explored by future research.

In conclusion, in 14- to 17-month-old infants, our data provide evidence for the parallel existence of episodic and semantic memories for the same experienced event, although these memories were not accessed simultaneously. Rather, episodic memory processing appears to suppress concurrent semantic processing—a mechanism that might be crucial for precise episodic retrieval and further semantic differentiation. As with the formation of infant semantic memories, the consolidation of infant episodic memories crucially depends on sleep after encoding. Here we found frontal fast sleep spindles to be involved in the sleep-dependent binding of episodic information. In concert with previous findings on semantic generalization[28–30], our results identify locally recruited sleep spindles as a key element of the "sleep-dependent memory evolution"[51] in early infancy.

## Methods

**Participants**. Data were obtained from 60 infants (26 female, mean age 15 months and 11 days, sd 30 days). An additional 47 infants (20 from the nap group, 27 from the wake group) were measured, but excluded from the analyses because of: too few artefact-free trials or very noisy ERP responses ($N = 34$), failure to fall asleep in the nap group (N = 5), fussiness, crying, or inattentiveness in one of the experimental sessions ($N = 4$), or due to technical problems with the acquisition of the sleep EEG ($N = 4$). Most infants participated in an additional learning study with artificial material, which was conducted on a different day with a temporal distance of about 7 days. All parents gave informed consent before participation. We have complied with all relevant ethical regulations. The study was approved by the ethics committee of Humboldt University of Berlin.

All infants were born in the 36th to 41st week of pregnancy with a birth weight ranging from 2100 to 4700 g (mean: $3512 \pm 520$ g). They had no known visual or hearing deficits and no major sleep problems. As typical for the investigated age group, all infants were habitual nappers. Prior to the lab visit, infants were assigned to either the wake group or the nap group. Infants of the nap group were scheduled at a time when they were expected to take a nap within the next hour. Infants of the wake group were scheduled at a time when they were expected not to take a nap within the next two to three hours. The nap group ($N = 30$, 11 female) and the wake group ($N = 30$, 15 female) did not significantly differ in age ($t_{59} = -1.561$, $P = 0.124$, t-test for independent samples), gestational age at birth ($t_{54} = -0.164$, $P = 0.871$), birth weight ($t_{54} = 0.076$, P = 0.940), and Apgar score (median of the APGAR score at 10 min after birth: 10 in both groups, Kolmogorov–Smirnov-Z = $-0.815$, $P = 0.415$).

**Procedure**. Infants participated in two experimental sessions, the encoding session and the memory test session, both conducted on the same day, with a retention period of about 0.5 to 2 h between them (Fig. 1). In the encoding session, infants were exposed to 64 pictures of individual objects, with each picture presented for 3200 ms. The pictured objects belonged to 8 different categories, and 8 different objects were shown for each category. After an interval of 800 ms post picture onset, an indefinite article was presented. After another 800 ms, when the object was still present on the screen, the correct word label for the object category was acoustically embedded into the context provided by the current picture. Each individual object–word pairing was presented twice. The encoding session was partitioned into blocks (of about 50 s), with the two presentations of an object–word pairing occurring always within a minute, i.e. either in the same block, or, when the first presentation was in the second half of a block, the repetition occurred in the first half of the subsequent block. The 8 initial trials (block 0 without repetitions), as well as the 8 last repetitions (block 8 without novels), were excluded from analyses. This block design ensured that analysed novel pairings and analysed repeated pairings were balanced over time (i.e., the same number of novel and repeated pairings were presented within the same time intervals), such that the analysis of repetition during encoding was not affected by perceptual or attentional changes over the course of the experimental session. There were no obligatory pauses between blocks, but short breaks were taken whenever necessary. Without breaks, the encoding session lasted for 7 min.

After encoding, infants of the nap group were prepared for polysomnographic recordings. Some of them were fed and freshly diapered before they were laid down in a baby crib, pram, or were held by their parent until they fell asleep. Preparation time before laying the infants down for sleep was about 10 min ($10.9 \pm 7.97$). Mean sleep onset latency from laying down was 24.1 min (SD 26.8). Subsequently, infants napped for about an hour ($57.0 \pm 21.9$ min). Compared to the retention time of the nap group ($108.7 \pm 36.1$ min), we shortened the retention time of the wake group ($40.8 \pm 6.9$ min) in order to ensure that infants stayed alert during the memory test. As a result, mean retention time in the wake group did not significantly differ from mean wake retention time before sleep onset in the nap group ($t_{58} = 1.293$, $P = 0.205$, t-test for independent samples). Also, the time of the day at which the memory test was applied ($13:04 \pm 2:16$ h) did not significantly differ between groups ($t_{58} = 1.575$, $P = .121$, wake group: $13:31 \pm 2:51$ h, nap group: $12:36 \pm 1.23$ h), since the encoding session was applied somewhat earlier in the nap than in the wake group (nap group: $10:50 \pm 1.15$ h, wake group: $12:51 \pm 2:52$ h for the end of the encoding session).

In the memory test, following the retention period, four old exemplars of each category, (i.e., objects that infants had already seen), and four novel exemplars, (i.e., objects that infants had not previously seen), were presented twice each. One presentation was paired with the correct word label and the other with an incorrect word label. Thus, the memory test phase had four conditions: semantically correct pairings with old objects (the same pairings as during encoding), semantically correct pairings with new objects, semantically incorrect pairings with old objects, and semantically incorrect pairings with new objects. Trials of these conditions were presented intermixed. None of the 128 object–word pairings (32 per condition) was repeated, whereas each individual object occurred twice and each word was presented 16 times. The memory test session was divided into 4 blocks (without obligatory pauses), with the two presentations of an object (for correct and incorrect pairings) occurring within the same block. The memory test session also lasted for 7 min.

**Stimuli**. Visual stimuli were 96 coloured images of real objects, which were isolated from digital photographs. Objects belonged to 8 basic-level categories, and 12 different objects were used for each category. Categories were car, ball, pail, dog, cookie, spoon, shoe, and bird. Auditory stimuli were 8 mono- or disyllabic German words that name the categories at the basic level (*Auto, Ball, Eimer, Hund, Keks, Löffel, Schuh, Vogel*). These words are known to be comprehended very early in life (mean comprehension in the standardized German parental questionnaire ELFRA–1[52] at 18 months: 92%). All words were stressed on the first syllable and in German are masculine or neuter. They were spoken slowly by a female speaker, digitized at a rate of 44.1 kHz, and presented through a loudspeaker with moderate intensity. Infant's comprehension of these words was assessed by parental ratings. The median of the rated number of comprehended words was 7 in both groups. Also, the frequency distribution of this number did not differ between groups (Kolmogorov–Smirnov-Z = 0.775, $P = 0.586$).

**EEG recording and ERP analysis**. During the encoding and memory test sessions, the EEG was recorded with a stationary system (REFA, Twente Medical Systems International) at 21 electrode sites and digitized on-line at a rate of 500 Hz. Off-line, the EEG was re-referenced to the average of left and right mastoids. A zero-phase digital band-pass filter ranging from 0.5 to 20 Hz (–3 dB cut-off frequencies at 0.61 and 19.89 Hz) was applied. The strong DC-suppression of this filter (–90 dB) enabled the calculation of ERPs without baseline correction[53]. ERPs were averaged from picture onset and analysed time-locked to word onset. Trials exceeding a standard deviation of 80 µV within a sliding window of 500 ms at any electrode site were rejected. Inattention during a trial is typically related to body movements. Hence, the rejection criterion was applied to the whole trial interval, from picture onset, to ensure that only trials in which infants had attended to the picture were included into ERP analyses. This kind of analysis resulted in an enhanced rejection rate, which led to the exclusion of more individuals from analyses. However, it also resulted in more robust effects than in previous infant studies with similar designs. A minimum of 7 artefact-free trials per condition was defined to be required for the inclusion of an individual in further analyses. On average 18 trials (SD = 4.47) per condition contributed to an individual participant's ERP. When analysing trial numbers by a repeated measures analysis of variance (ANOVA) with the within-subject factors Old/New (old object context vs. new object context) and Semantic Congruity (semantically correct pairing vs. semantically incorrect pairing) and the between-subject factor Group (wake vs. nap), an interaction between Old/New and Group ($F_{1,58} = 6.048$, $P = 0.017$ $\eta_p^2 = 0.094$) indicated specific differences in trial numbers between groups. Post-hoc analyses revealed that, in the nap group, the mean number of trials in the new object context was slightly higher than that in old object context (19.75 vs. 18.65, $t_{1,29} = -2.241$, $P = .033$, $d = -0.242$, paired t-test). As an estimation of attention, this enhanced trial number may be seen as novelty preference for trials with new objects compared to trials with old objects. This provides behavioural evidence for the retention of specific episodic information in infants who had napped during the retention period. A similar preference was not observed in infants who had stayed awake ($t_{1,29} = 1.213$, $P = 0.235$, paired t-test; wake–nap group difference for new

objects: $t_{2,58} = -2.598$, $P = 0.012$, $d = -0.640$). Further, the number of trials with old object context did not differ between groups ($t_{1,58} = -1.172$, $P = 0.246$, t-test for independent samples).

In order to increase statistical power and to assess ERP effects with different spatial extent in parallel, regions of interest (ROIs) were defined by averaging ERP responses of individual recording sites. F7 and T7 formed the left fronto-temporal region (LFT), F8 and T8 the right fronto-temporal region (RFT), CP5 and P7 the left parietal region (LP), and CP6 and P8 the right parietal region (RP). Similarly, F3, FZ, F4, C3, CZ, and C4 were included into the mid-fronto-central region (mid FC), and P3, PZ, P4, O1, and O2 into the mid-parieto-occipital region (mid PO). ROIs and the analysed time window from 400 to 800 ms were chosen on the basis of previous infant studies[28,29,33,37,43] and of visual inspection of the data. For better illustration of the ERP effects, an additional low-pass filter of 7 Hz was applied to the averaged ERPs shown in Figs. 2 and 3, as well as in Supplementary Figs. 1 and 2.

To evaluate episodic and semantic context memory and its dependency on sleep, we conducted an overall ANOVA on the ERP data of the memory test. The within-subject factors included: Old/New (old object context vs. new object context), Semantic Congruity (semantically correct pairing vs. semantically incorrect pairing), Laterality (left, mid, right), and Region (anterior vs. posterior) and the between-subject factor Group (wake vs. nap). After significance was revealed for the Old/New × Semantic Congruity × Group interaction, the ERP differences between words in incorrect pairings and words in correct pairings were separately analysed for old and new object contexts in the two groups. For testing ERP differences at six individual ROIs (due to the Semantic Congruity × Laterality interaction and the Semantic Congruity × Region interaction), a conservatively corrected significance level of .008 was chosen. When testing differences within groups, Cohen's d was calculated as the quotient of the mean and the standard deviation of the differences. When applying t-tests for independent samples, Cohen's d was calculated as the quotient of the mean group difference and the pooled standard deviation. For group comparisons, visualizations, and correlations, we combined the significant effects at mid-parieto-occipital, left parietal, and right parietal ROIs by calculating the overall N400 effect over the parieto-occipital (PO) region as mean of the effects at the positions P3, PZ, P4, CP5, CP6, P7, P8, O1, and O2.

In order to test whether the episodic memory effect observed over the LFT region in the nap group was based on the immediate formation of episodic memories during encoding, we analysed the data of the encoding session with an ANOVA with the within-subject factors Repetition (repeated object context vs. novel object context), Laterality, and Region. We specifically tested for an effect of Repetition over the LFT region. A correlation analysis was also performed between the (non-significant) repetition-related FTMR-like ERP difference at encoding and the episodic old/new FTMR effect observed in the memory test.

To investigate object memory, we calculated the ERP responses time-locked to picture-onset for two time windows (early: 150–300 ms, late: 600–800 ms). We specifically analysed the responses to the first presentation of an object in the memory test (independent of whether it was followed by a correct or an incorrect word) with ANOVAs. These ANOVAs featured the within subject factors Old/New (old vs. new object) and Region (frontal, central, parietal, and occipital) and the between-subject factor Group (wake vs. nap). The immediate object memory formed during encoding was tested by an ANOVA with the factors Repetition (repeated vs. novel object) and Region. The frontal region was defined as the mean of the ERP amplitudes at F3, FZ, and F4, the central region included FC3, FC4, C3, CZ, and C4, the parietal region P3, PZ, and P4, and the occipital region O1 and O2. For testing ERP differences at the four individual regions, the significance level was adjusted to .0125.

All statistical tests were two-sided. In all ANOVAS, Greenhouse–Geisser-adjusted P-values were reported whenever degrees of freedom were >1.

**Sleep recordings and sleep spindle analyses**. Sleep was recorded using a portable amplifier (SOMNOscreen EEG 10–20, Somnomedics, Kist, Germany). EEG recordings were obtained with electrodes attached at F3, FZ, F4, C3, C4, P3, PZ, P4, left and right mastoids, referenced to Cz (positions according to the International 10–20 system), filtered between 0.03 and 35 Hz, and sampled at 256 Hz. The electrooculogram and the electromyogram were recorded bipolar from electrodes close to the eyes and at the chin, respectively. Offline, EEG signals were re-referenced to the average potential at left and right mastoid electrodes. Sleep recordings were visually scored according to standard criteria[54–56]. For each nap, total sleep time and the time spent in the different sleep stages (1, 2, slow wave sleep, and REM sleep) were determined.

The algorithm for the detection of discrete sleep spindles was adopted from Mölle et al.[25]. First, for each infant and each channel, the individual peak frequency of fast sleep spindles ($13.93 \pm 0.17$ Hz, across all infants and channels) was identified in the EEG power spectra of the low-pass filtered (32 Hz) and down-sampled (128 Hz) EEG of all artefact free NonREM epochs. The EEG signal was then filtered with a band-pass width of 3 Hz centred on the detected individual peak frequency. A root mean-square (RMS) representation of the filtered signal was calculated and smoothed using a sliding window of 0.2 s with a step size of one sample. Time frames were considered as spindle intervals if the RMS signal exceeded a threshold of 1.5 standard deviations of the filtered signal ($5.99 \pm 1.32$ μV, across all infants and channels) for 0.5–5 s ($1.056 \pm 0.096$ s) and if

the largest value within the frame was greater than 2.5 standard deviations of the filtered signal ($9.97 \pm 2.19$ μV). Two succeeding spindles were counted as one spindle when the interval between the end of the first spindle and the beginning of the second spindle was shorter than 0.5 s and the resulting (merged) spindle was not longer than 5 s. For each infant, the mean RMS amplitude of fast spindles was calculated for each channel by averaging the individual spindle RMS amplitudes. RMS amplitude of an individual spindle was calculated by summing the squares of all amplitude values of the band-pass filtered signal from the beginning to the end of the spindle, dividing the sum by the number of values, and taking the square root. Pearson's correlation coefficients between the FTMR in the ERP of the memory test and spindle RMS amplitude at six channels (F3, F4, C3, C4, P3, and P4) were calculated and tested with an adjusted significance level of .008. In additional explorative analyses, we calculated the correlation coefficients between the ERP effects and other spindle parameters. While correlations with peak-to-peak amplitude yielded very similar results as those reported for RMS amplitude, correlations with spindle length, spindle number, and spindle density did not reach significance and were not reported.

**Reporting summary**. Further information on research design is available in the Nature Research Reporting Summary linked to this article.

## Data availability
All relevant data are available upon reasonable request. Inquiries should be directed to the corresponding author. The source data underlying Figs. 2a–c, 3a–d, and 4b and Supplementary Figs. 1a–c and 2a–d are provided as a Source Data file. A reporting summary for this article is available as a Supplementary Information file.

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

## Acknowledgements

We thank all families who participated in this study. Special thanks to Christina Rügen for recording the infant ERP data and to Kerstin Strelow-Morgenstern for recruiting participants and scoring the infant sleep data. The study was supported by grants from the Deutsche Forschungsgemeinschaft to M.F. (FR 1336/2–1, FR 1336/2–2, and FR 1336/3–1).

## Author contributions

Conceptualization, M.F.; analysis, M.F. and M.M.; writing original draft, M.F.; writing review and editing, M.M., A.D.F., and J.B.; resources, M.F. and A.D.F.; funding acquisition, M.F.

## Competing interests

The authors declare no competing interests.
