## [Peer Review File · Nature Communications]

Reviewers' Comments:

Reviewer #1:

Remarks to the Author:

This aim of this study was to explore whether sleep in 14- to 17-month-old infants support the consolidation of individual object-word pairings as specific episodic-like memories, and whether sleep spindles are involved in this consolidation process. A secondary aim was to explore the interaction of related episodic and semantic memories during their earliest stages. Participants were shown object-label pairings of 8 familiar basic-level categories (e.g. a picture of a dog) and its auditory label (e.g. 'hund' in German) repeatedly during the encoding stage. Subsequently, half of the participants took a nap, while the other half stayed awake for similar amount of time. Finally, all participants took part in a memory test where there was a 2 (Old vs. New Object) x 2 (Correct vs. Incorrect Label Pairing) design. The old object is the same item participants saw in the encoding stage (e.g. the same dog), while the new object is a different exemplar if the same category (e.g. a different dog).

The authors reported two main findings. Firstly, the N400 effect was not observed in the nap group when the old object was paired with an incorrect label. As seen in Figure 2a in the manuscript, the N400 effect was observed in the comparison of the correctly- vs. incorrectly-labelled new objects, in both the nap and wake groups, reflecting the detection of semantic mismatch of the object and label. As seen in Figure 2b, the wake group also showed an N400 effect with the old objects. However, as seen in Figure 2c, the nap group showed what the authors termed the FTMR episodic context effect without concurrent N400 effects. That is, the nap group participants did not 'detect' the semantic mismatch when the object objects was paired with the incorrect label (e.g. the picture of the dog with the label 'auto (car)'). Secondly, the amplitude of the left FTMR was correlated with the RMS amplitude of frontal fast sleep spindles in the nap group participants. The authors concluded that their findings indicate that sleep supports episodic memory consolidation in infants, and that semantic processing of an item is temporally suppressed if the item has recently been involved in the formation of episodic memory (in this case the specific object-label association).

Comments

The findings of the current study are interesting and potentially important to the community. However, the finding of semantic processing suppression seems to be an incidental finding, which is novel and counterintuitive, the current manuscript does not provide sufficient detail information and transparent data representation for the reviewer to assess whether the study was performed to a high technical standard, and whether the data supports the authors' claim.

The methodology and data analysis are unclear:

a) Given the large amount of participant exclusion, the authors should indicate the distribution of these excluded participants in the wake vs. nap group.

b) While there is more than a page of description of the design of the encoding stage (which is not analysed extensively), there is less than half a page of description of the memory test session (the main analysis). It is unclear how many trials were participants shown in the test session (as opposed to only reporting how many trials were artefact free for analysis). This is important because the findings could be influenced by the amount of repetition. Could the nap group be still

tired after the nap?

c) Sleep architecture of participants is not reported.

d) For the reviewers and in the appendix, data of individual channels and the distribution of the N400 effects of the nap vs. wake groups should be visualised, in support of such novel finding, demonstrating that the effects were not resulted from data noise and outliers.

e) Has the author looked at phonological mismatch at the onset of the target word? Given the incorrect label has a different onset.

The authors made little to no attempt to provide theoretical support or discussion of the finding in the introduction /discussion. Particularly, what does this FTMR reflect? The authors mentioned that the FTMR shares spatial characteristics with the N200-500 infant ERP response to words. A similarly-designed study published by the current authors in Nature Communications in 2015, reported an N200-500 effect without concurrent N400 effect in a 'specific word meaning' condition, in which participants saw correct or incorrect pairings of old object-label combinations. There is scope for further elaboration and discussion of the findings of the current study.

Reviewer #2:

Remarks to the Author:

The present study seeks to examine the formation of episodic memories in 14- to 17-month-olds. This goal is pursued by comparing ERP components as a function of whether infants are processing correct versus incorrect word-pair associations and whether they have or have not had a nap. The study has many significant strengths and a few weaknesses that I will list below.

Strengths

- The study addresses an important question. Researchers have long asked about how early memories may be retained as either episodic or semantic representations and this study is designed to examine how the brain might dynamically engage in both processes at the formation of new memories
- The study utilizes a sleep manipulation to gain insight on the status of newly formed memory based on previous evidence of differential effects of sleep versus wakefulness on memory specificity and generalization.
- The results are interesting, clear, and novel. Infants view objects and hear their labels. After an interval, all infants show a N400 if one of the viewed objects is paired with a wrong label. However, only after wakefulness, infants show a N400 if a new exemplar of a viewed object (i.e., a different looking car) is paired with the label. IF they napped, they show a different component that is interpreted as a signature of episodic memory. The magnitude of this component is correlated with spindles during NREM sleep.

Weaknesses

- Although the ERP results are interesting, the absence of a behavioral measure of memory precludes a direct connection of the ERP findings to what infants can demonstrate to have learned. There is evidence that more incorrect pairing trials are retained in the analysis, which the authors suggest might have to do with their perceived novelty, but this is not sufficient to determine infants' retention or whether retention differs between nap and wake groups (given that they are tested at different intervals).
- The interpretation of the results is overreaching at times. For example, the new component is interpreted as being a sign of episodic memory, but it could also correspond to a memory updating process or a familiarity response. The absence of an N400 is interpreted as inhibition of semantic memory. For example, it could indicate an attentional modulation toward the recent memory. Regardless, absence of N400 alone does not indicate the presence of an inhibitory process.

Additional evidence would be necessary to make this claim. In addition, I am not clear as to why wakefulness would favor semantic processing.

- Episodic memory is typically indicated by retention of novel and arbitrary item-context associations, but word-objects associations are already known in this experiment. At such, it seems more prudent to characterize these memories as declarative reflecting familiarity processes.

Reviewer #3:

Remarks to the Author:

This study used ERPs to track episodic and semantic memory effects in 14- to 17-month old infants, with either a period of wake or sleep occurring between a learning and test session. The authors used an elegant task design comprising semantically correct and incorrect object-word pairings, with both new and old objects used in the test phase (while holding the word constant). The authors report that:

- i) Infants show an N400 response for words presented in a semantically-incongruent context (new objects, correct vs. incorrect pairings). This response did not differ between nap and wake groups.
- ii) For old objects, the N400 response (old objects, correct vs. incorrect pairings) was only present in the wake group, not the nap group. In the wake group, the N400 effect for old and new contexts was not different.
- iii) However, there was no N400 response for infants who napped (old objects, correct vs. incorrect pairings). Instead, these infants showed a memory response (FTMR), characterized by increased negativity in response to old, correct pairings (compared to old, incorrect and new, correct). This memory response was only evident in the test session following the nap, not during the repeated trials during encoding. Further, the episodic memory response was correlated with spindle amplitude during the nap.

These findings are novel and advance our understanding of sleep-dependent consolidation in infants, a difficult population to study. The methods appear to be sound and the statistical analyses are appropriate. I have a few questions and concerns:

1. I would be interested in seeing more of the data presented in the Figures (and maybe a Table).

My suggestions:

- a. Report the time of day means of the encoding and test sessions in each group individually.
- b. Show the lack of FTMR for the Wake group in Figure 2.
- c. Show the encoding and memory LFT plots for the Wake group in Figure 3.
- d. Report the nap sleep descriptives.

2. Is the duration of the retention interval correlated with the memory outcome?

3. Was spindle number or spindle density correlated with FTMR? I'm assuming they were not since they are not reported here. However, I think it is informative to report this (even if non-significant) since many people in the field use these metrics, and it is useful to keep track of these effects. Similarly, were any other sleep variables correlated with outcomes?

4. Can the authors speculate about why the episodic memory effect seems to be specific to hearing the word following the object, and there doesn't seem to be an old vs. new recognition response following picture onset.

5. There are a couple instances where the authors present their post-hoc interpretation of the results almost as fact (e.g., on page 10 in the Discussion: "The answer is that newly formed episodic memories..."). I suggest some of this language be reworded to be more speculative in nature.

6. Minor points:

- a. Page 3, Introduction: "A still unsolved question is, on which factors..." - I find this sentence to be unclear and slightly confusing.
- b. In Figures 2 and 3, I suggest listing all the electrodes included in PO and LFT to the figure caption.
- c. Effect sizes should be reported.

REVIEWER COMMENTS:

Reviewer #1:

General Comments

(1) The findings of the current study are interesting and potentially important to the community. However, the finding of semantic processing suppression seems to be an incidental finding, which is novel and counterintuitive, the current manuscript does not provide sufficient detail information and transparent data representation for the reviewer to assess whether the study was performed to a high technical standard, and whether the data supports the authors' claim.

Authors' response: Although the observed semantic processing suppression appears to be somewhat unexpected, it was not an incidental finding. As the reviewer her/himself points out in her/his note #8, in the nap group of a previous study, we found a related pattern for initially unknown objects and words: an N200-500 effect without concurrent N400 in the specific word meaning condition, in which the same objects as in the learning phase were presented in the memory test, while novel objects of the general word meaning condition elicited an N400 effect (Friedrich et al., Nature Communications, 2015). In fact, this previous finding for the specific word meaning condition was one of the main motivations of the current study. However, since we are aiming at a broader readership, we were hesitant to refer to this specific motivation in the Introduction. Instead, we preferred to use the more general phrasing "interrelation between episodic and semantic memories" in order to refer to the question, whether and how early episodic memory affects infant semantic memory processing.

In the discussion, however, we have related our novel findings to that of the previous study. Please see our detailed response to note #8.

For detail information and transparent data representation, see our responses to the specific comments.

Specific Comments

(2) Given the large amount of participant exclusion, the authors should indicate the distribution of these excluded participants in the wake vs. nap group.

Authors' response: In the method section we have now specified: "An additional 47 infants (20 of the nap group, 27 of the wake group) were measured, but excluded from the analyses" (p22) The somewhat higher exclusion rate in the wake group was due to the fact that after the wake period more infants appeared to be less attentive and more restless than after the nap (which more often produced artifacts and, thus, resulted in too few trials for ERP averaging).

(3) While there is more than a page of description of the design of the encoding stage (which is not analysed extensively), there is less than half a page of description of the memory test session (the main analysis). It is unclear how many trials were participants shown in the test session (as opposed to only reporting how many trials were artefact free for analysis). This is important because the findings could be influenced by the amount of repetition.

Authors' response: We apologize for the missing information. We have now added the relevant information: "None of the 128 object–word pairs (32 per condition) were repeated, whereas each individual object occurred twice and each word was presented 16 times." (p24)

(4) Could the nap group be still tired after the nap?

Authors' response: This is rather unlikely, because we applied the memory test not until an infant was wide awake. Also, the brain response to words in novel object context did not differ between infants of the nap and the wake group.

(5) Sleep architecture of participants is not reported.

Authors' response: We have now included a table with the sleep characteristics.

Tab. 1

	Mean	SD
Stage 1 sleep	11.55	7.76
Stage 2 sleep	25.81	12.40
Slow wave sleep	19.67	10.70
REM sleep	0.00	0.00
TST	57.02	21.86
Spindle RMS frontal	11.27	3.08
Spindle RMS central-parietal	9.19	2.11

(6) For the reviewers and in the appendix, data of individual channels and the distribution of the N400 effects of the nap vs. wake groups should be visualised, in support of such novel finding, demonstrating that the effects were not resulted from data noise and outliers.

Authors' response: In response to the reviewer's comment, for each of the panels in figure 2 we have included a corresponding figure into the supplemental material (Fig. S1a, b, and c), showing the distribution of the effects at individual channels. These figures demonstrate that the effects do not result from data noise.

Also, the frequency distributions of the mean ERP amplitude in the N400 range (each fitted into an appropriate normal distribution, see below) shows that neither the observed effects

nor the missing effect result from outliers. Indeed, there were no extreme outliers in the data.

Wake group, new:

Nap group, new:

Wake group, old:

Nap group, old:

(7) Has the author looked at phonological mismatch at the onset of the target word? Given the incorrect label has a different onset.

Authors' response: In the “Incorrect” condition, there was always a phonological mismatch at the onset of a target word. However, the present study was not designed to investigate phonological mismatch. In particular, the high number of individual word repetitions is expected to attenuate such an effect. Nevertheless, since the FTMR shares characteristics with the N200–500, we discussed the possible impact of episodic memory on phonological word processing:

“...The N200–500 reflects a non-semantic word processing stage that is affected by stimulus repetition, familiarity, and expectation. Most likely it is linked to an acoustic-phonological stage of processing such as the activation of acoustic-phonological features or their binding to a coherent word. If the FTMR represents an N200–500 response, this will imply that newly formed episodic memory representations for the specific pairings of objects and words facilitate acoustic-phonological word processing, when a word is re-experienced in its same episodic object context. A word in the context of an old object would then be processed as it would be more familiar than the same word in the context of a new object. Whether the FTMR indeed reflects such a modality-specific effect of episodic memory on word processing, i.e., a kind of perceptual priming, or rather a modality-independent effect, such as the facilitated integration of the correct item into the current episode, which would represent a kind of “episodic priming”, remains to be solved by future studies.” (p10/11)

(8) The authors made little to no attempt to provide theoretical support or discussion of the finding in the introduction /discussion. Particularly, what does this FTMR reflect? The authors mentioned that the FTMR shares spatial characteristics with the N200-500 infant ERP response to words. A similarly-designed study published by the current authors in Nature Communications in 2015, reported an N200-500 effect without concurrent N400 effect in a 'specific word meaning' condition, in which participants saw correct or incorrect pairings of old object-label combinations. There is scope for further elaboration and discussion of the findings of the current study.

Authors' response: In response to the reviewer’s note, we have added a paragraph, in which we have discussed this issue in more detail:

“A similar pattern of ERP effects as in the present study has been found in the nap group of a learning study, in which 9- to 16-month-old infants were presented with pairings of objects and words, for which they had no lexical-semantic pre-knowledge²⁸. For pairings with novel objects in a general (category) learning condition, which the infants had not experienced before the nap, an N400 effect emerged in the memory test after the nap. For pairings with individual objects, to which the infants had been repeatedly exposed in the specific learning condition before the nap, an N200–500 effect without concurrent N400 was observed. The similarity of this N200–500 with the here observed FTMR suggests that, as in the present study, episodic memories were built in the specific learning condition of that previous study. In particular, the missing object variability in the specific learning condition could have triggered the formation of episodic instead of semantic memories, while in the general learning condition with high object variability, new semantic memories were built. Since object variability in the present study was quite similar to that of the general learning condition in the previous study, here, the pre-existing lexical-semantic memory appears to be the crucial factor that led to the consolidation of episodic memories. It still remains open, whether the consolidation of a new semantic memory precludes the consolidation of a related episodic memory or whether these memories could even be built in parallel.” (p 11/12)

Reviewer #2

The present study seeks to examine the formation of episodic memories in 14- to 17-month-olds. This goal is pursued by comparing ERP components as a function of whether infants are processing correct versus incorrect word-pair associations and whether they have or have not had a nap. The study has many significant strengths and a few weaknesses that I will list below.

Strengths

- *The study addresses an important question. Researchers have long asked about how early memories may be retained as either episodic or semantic representations and this study is designed to examine how the brain might dynamically engage in both processes at the formation of new memories*
- *The study utilizes a sleep manipulation to gain insight on the status of newly formed memory based on previous evidence of differential effects of sleep versus wakefulness on memory specificity and generalization.*
- *The results are interesting, clear, and novel. Infants view objects and hear their labels. After an interval, all infants show a N400 if one of the viewed objects is paired with a wrong label. However, only after wakefulness, infants show a N400 if a new exemplar of a viewed object (i.e., a different looking car) is paired with the label. IF they napped, they show a different component that is interpreted as a signature of episodic memory. The magnitude of this component is correlated with spindles during NREM sleep.*

Authors' response: We thank the reviewer very much for her/his appreciation.

Weaknesses

(1) Although the ERP results are interesting, the absence of a behavioral measure of memory precludes a direct connection of the ERP findings to what infants can demonstrate to have learned. There is evidence that more incorrect pairing trials are retained in the analysis, which the authors suggest might have to do with their perceived novelty, but this is not sufficient to determine infants' retention or whether retention differs between nap and wake groups (given that they are tested at different intervals).

Authors' response: While we respect the reviewer's concern, we want to pinpoint that to date there is no evidence that any of the behavioral measures commonly applied in infants in the investigated age range provides more valid markers of the memory processes of interest than brain responses assessed by ERPs.

Rather, behavioral measures such as exploring preference or looking time measures are indirect measures that rely on the modulation of an infant's attention. These measures involve an interference with the developmental trajectory of the attentional response, e.g. the shift from familiarity preference to novelty preference, which not only depends on brain maturation, but is also non-linearly affected by experimental factors such as familiarization time (Houston-Price & Nakai, 2004). The strength of the relevant memory representations might thus affect an infant's attention in opposite ways (first positive, later negative), which very much complicates the interpretation of behavioral data in a multifactor-design.

More importantly, even when using a direct method such as pointing at the named object, the interrelation of episodic and semantic memory processing cannot not be assessed, since behavioral measures do not enable the distinction between knowledge based on episodic memory and knowledge based on semantic memory. That means, behaviorally, it cannot be specified, whether an old pairing is recognized as episode or as lexical-semantic item.

In contrast, the N400 is a well-established and extremely well-investigated ERP component that most probably, if not certainly, indicates a semantic processing stage in adults (Kutas & Federmeier, 2011). When applying the very same experimental designs to infants and adults, similar N400 effects have been observed in both groups (Friedrich & Friederici, JoCN 2005; Friedrich & Friederici, NeuroReport 2005). Since then, N400 effects in infants and toddlers have been reported by several research groups in a number of different designs, for object-word pairings, word-object pairings, word-word pairings, and sentence processing (e.g., Parise & Csibra, 2012; Torkildsen et al., 2007; Råma et al., 2013; Friedrich & Friederici, 2005). The amplitude of the N400 is moreover related to an infant's behavioral language development, i.e. to the infant's individual comprehension and speech production abilities (e.g., Friedrich & Friederici, 2006; Torkildsen et al., 2008; Friedrich & Friederici, 2010; Junge et al., 2012). On the whole, there is ample evidence for the view that the infant N400 indexes the presence of early semantic knowledge. Thus, even without behavioral measures, semantic memory can be attested in early infancy.

It would, of course, be interesting to see, whether and how the kind of memory use (episodic vs. semantic) would be reflected in an infant's attention, as well as to relate individual ERP components (N400, FTMR) to changes reflected in the infant behavior. This is, however, a completely different research topic and a challenging task, which would require to develop experimental settings that are suitable for the parallel assessment of behavioral and brain measures in infants.

By utilizing the number of artefact-free trials, we offer a first approach to the complex topic. We analyzed this available number as rough estimation of an infant's attention, while being aware that it is not an established and well-controlled experimental variable. Therefore, we reported it as supporting result in the method section, which may be particularly interesting for researchers in the developmental domain, but did not refer to it in the result section or the discussion.

(2) The interpretation of the results is overreaching at times. For example, the new component is interpreted as being a sign of episodic memory, but it could also correspond to a memory updating process or a familiarity response. The absence of an N400 is interpreted as inhibition of semantic memory. For example, it could indicate an attentional modulation toward the recent memory. Regardless, absence of N400 alone does not indicate the presence of an inhibitory process. Additional evidence would be necessary to make this claim. In addition, I am not clear as to why wakefulness would favor semantic processing.

Authors' response: Indeed, at present we do not know, what kind of processing the newly observed FTMR component reflects. As the reviewer argues, it might correspond to a memory updating process or an attentional response to the recent memory (which both implies the existence of recent memory for the episodic pairings) as well as it might represent an episodic binding effect or a familiarity response as we discussed at pages 10/11. However, whatever stage of processing the FTMR reflects, it indicates that the specific old object context of a word affects the processing of that word. This result clearly evidences the presence of memory for the episodic context, in which the word was presented.

Also, we are aware that the absence of an N400 alone does not indicate the presence of an inhibitory process. However, the absence of the N400 in response to words presented with old objects in conjunction with the presence of an N400 in response to the very same words presented with new objects, both observed in the same infants (in concert with the presence an N400 effect to both words presented with old and with new objects in the group of wake infants of the same age), provides strong evidence that semantic memory is present in the infants of the nap group, but that episodic memory specifically interferes with access to these memories.

This interpretation does not imply that wakefulness generally favors semantic processing. Rather, the pattern of results suggest that, in the wake group, episodic memory has not been retained and therefore it does not affect subsequent semantic processing.

Our interpretation is now further supported by the data of the picture processing, which we have newly analyzed and presented in the Result section (p8/9) and Fig. S2. Although the new findings show that post-encoding sleep facilitates not only memory for the episodic binding of objects and words, but also recognition memory for the objects, they speak for the notion that “recognition memory for a specific object and episodic memory for the pairing of a specific objects with a word are supported by distinct consolidation mechanisms during infant sleep.” (p 9, please see also our response to comment #4 by Reviewer #3).

In response to the reviewer’s note, we have carefully revised the manuscript and reworded several phrases in order to tone down our interpretations. In particular, we changed the sentence “The answer is that newly formed episodic memories in these infants interfered with semantic memories...” into: “The most probable explanation is that the newly formed episodic memories in these infants interfered with semantic memories...” (p13).

(3) Episodic memory is typically indicated by retention of novel and arbitrary item-context associations, but word-objects associations are already known in this experiment. At such, it seems more prudent to characterize these memories as declarative reflecting familiarity processes.

Authors’ response: The present study was specifically designed to uncover the relation between episodic and semantic memory by accessing the impact of specific item–context associations on pre-existing general knowledge. In our opinion, pairings with novel objects of known categories are particularly suitable to study this relation in infants, even though this kind of stimulus material is not yet established in research on episodic memory in adults.

The sleep-dependent retention of novel and arbitrary item-context associations in infants (without any pre-knowledge) has been investigated in a previous learning study. In response to the reviewer’s note, we have now discussed the relation between this and the current study, in particular with respect to the aspect of pre-knowledge:

“A similar pattern of ERP effects as in the present study has been found in the nap group of a learning study, in which 9- to 16-month-old infants were presented with pairings of objects and words, for which they had no lexical-semantic pre-knowledge²⁸. For pairings with novel objects in a general (category) learning condition, which the infants had not experienced before the nap, an N400 effect emerged in the memory test after the nap. For pairings with individual objects, to which the infants had been repeatedly exposed in the specific learning condition before the nap, an N200–500 effect without concurrent N400 was observed. The similarity of this N200–500 with the here observed FTMR suggests that, as in the present study, episodic memories were built in the specific learning condition of that previous study. In particular, the missing object variability in the specific learning condition could have triggered the formation of episodic instead of semantic memories, while in the general learning condition with high object variability, new semantic memories were built. Since object variability in the present study was quite similar to that of the general learning condition in the previous study, here, the pre-existing lexical-semantic memory appears to be the crucial factor that led to the consolidation of episodic memories. It still remains open,

whether the consolidation of a new semantic memory precludes the consolidation of a related episodic memory or whether these memories could even be built in parallel.” (p11/12)

Moreover, like the reviewer, we favor the view that the FTMR reflects a kind of familiarity response that is stronger when a word occurs in the context of a recently experienced object than when it occurs in the context of a novel object. In response to the reviewer’s comment, we have now referred to familiarity more explicitly, when discussing this issue:

“...If the FTMR represents an N200–500 response, this will imply that newly formed episodic memory representations for the specific pairings of objects and words facilitate acoustic-phonological word processing, when a word is re-experienced in its same episodic object context. A word in the context of an old object would then be processed as it would be more familiar than the same word in the context of a new object.” (11)

Reviewer #3

This study used ERPs to track episodic and semantic memory effects in 14- to 17-month old infants, with either a period of wake or sleep occurring between a learning and test session. The authors used an elegant task design comprising semantically correct and incorrect object-word pairings, with both new and old objects used in the test phase (while holding the word constant). The authors report that:

- i) Infants show an N400 response for words presented in a semantically-incongruent context (new objects, correct vs. incorrect pairings). This response did not differ between nap and wake groups.*
- ii) For old objects, the N400 response (old objects, correct vs. incorrect pairings) was only present in the wake group, not the nap group. In the wake group, the N400 effect for old and new contexts was not different.*
- iii) However, there was no N400 response for infants who napped (old objects, correct vs. incorrect pairings). Instead, these infants showed a memory response (FTMR), characterized by increased negativity in response to old, correct pairings (compared to old, incorrect and new, correct). This memory response was only evident in the test session following the nap, not during the repeated trials during encoding. Further, the episodic memory response was correlated with spindle amplitude during the nap.*

These findings are novel and advance our understanding of sleep-dependent consolidation in infants, a difficult population to study. The methods appear to be sound and the statistical analyses are appropriate. I have a few questions and concerns:

(1) I would be interested in seeing more of the data presented in the Figures (and maybe a Table). My suggestions:

- a. Report the time of day means of the encoding and test sessions in each group individually.*
- b. Show the lack of FTMR for the Wake group in Figure 2.*

c. Show the encoding and memory LFT plots for the Wake group in Figure 3.

d. Report the nap sleep descriptives.

Authors' responses:

a. In the Method section, we have now specified: "Also, the time of the day at which the memory test was applied ($13:04 \pm 2:16$ h) did not significantly differ between groups ($t_{58} = 1.575$, $P = .121$, wake group: $13:31 \pm 2:51$ h, nap group: $12:36 \pm 1.23$ h), which resulted from the fact that the encoding session was applied somewhat earlier in the nap than in the wake group (nap group: $10:50 \pm 1.15$ h, wake group: $12:51 \pm 2:52$ h for the end of the encoding session)." (p24)

b. We have included the LFT ROI of the wake group into Figure 2b, which shows the lack of FTMR in this group.

c. Also, we have included encoding and memory LFT plots for the Wake group in Figure 3.

d. The nap sleep descriptives are now shown in the new Table 1.

(2) Is the duration of the retention interval correlated with the memory outcome?

Authors' response:

For the N400 we have now stated: "Also, it was not correlated with the duration of the retention interval, neither in the nap group ($r = .123$, $P = .519$) nor in the wake group ($r = .082$, $P = .665$)." (p6)

For the FTMR, we have included at page 7: "Also, it was neither correlated with the duration of the retention interval ($r = .098$, $P = .605$)..."

(3) Was spindle number or spindle density correlated with FTMR? I'm assuming they were not since they are not reported here. However, I think it is informative to report this (even if non-significant) since many people in the field use these metrics, and it is useful to keep track of these effects. Similarly, were any other sleep variables correlated with outcomes?

Authors' response:

For the FTMR we stated in the Result section: "Also, it was neither correlated with ... nor with total sleep time ($r = -.182$, $P = .337$) or the time spent in individual sleep stages ($|r| = .060 - .204$, $P = .280 - .753$)." (p7)

We agree with the reviewer that information about the non-significant correlations with other specific spindle parameters will be informative for scientist in the field of sleep research. However, we would not present this rather specific information in the main text of a journal with a broad readership. Instead, we included a statement in the method section (p26):

"In additional explorative analyses, we calculated the correlation coefficients between the ERP effects and other spindle parameters. While correlations with peak-to-peak amplitude

yield very similar results as those reported for RMS amplitude, correlations with spindle length, spindle number, and spindle density did not reach significance and were not reported.” (p29)

(4) Can the authors speculate about why the episodic memory effect seems to be specific to hearing the word following the object, and there doesn't seem to be an old vs. new recognition response following picture onset.

Authors' response: This is, indeed, an interesting point. When comparing the responses of all old vs. all new pictured objects, there is no old-new recognition effect in the ERP. In the memory test session, however, each object was presented twice, once in the semantically correct condition and once in semantically incorrect condition. In response to the reviewer's comment, now we have restricted the analysis of the old-new effect to the first presentation of an individual object in the memory test, i.e., to those trials, in which an old object was not yet repeated and in which a new object was indeed novel. This additionally analysis revealed a distinct pattern. We therefore included a new paragraph in the Result section:

“In order to explore, how this memory for the episodic pairing of objects and words was related to recognition memory for the specific objects, we analyzed the brain responses to the first presentation of an individual object in the memory test. An early-latency old/new memory effect over the occipital region (Old/New \times Region $F_{3,174} = 8.338$, $P = .001$, $\eta_p^2 = .126$; occipital $t_{59} = -3.505$, $P = .001$, $d = -.452$; Fig. S2b upper panel) indicated visual memory for the specific objects in infants of both groups (wake: $t_{29} = -2.489$, $P = .019$, $d = -.454$, nap: $t_{29} = -2.678$, $P = .012$, $d = -.489$, group comparison: $t_{58} = -.817$, $P = .417$). A second old/new memory effect over frontal and central regions at a latency of 600 – 800 ms, which most-likely indicates higher-level recognition memory, was present only in the nap group (Old/New \times Wake/Nap $F_{1,58} = 10.276$, $P = .002$, $\eta_p^2 = .151$, Old/New \times Region $F_{3,174} = 4.294$, $P = .014$, $\eta_p^2 = .069$, nap frontal: $t_{29} = -3.368$, $P = .002$, $d = -.615$, nap central: $t_{29} = -3.733$, $P = .001$, $d = -.682$, wake frontal: $t_{29} = 1.382$, $P = .178$, wake central: $t_{29} = 1.299$, $P = .204$, group difference for the overall frontal-central effect: $t_{58} = 3.538$, $P = .001$, $d = .632$; Fig. S2b upper panel, Fig. S2d). None of these old/new effects in response to the pictured objects was correlated with the FTMR episodic context effect in response to words (early occipital: $r = .168$, $P = .199$, late frontal-central, nap: $r = -.102$, $P = .440$, late frontal-central, wake: $r = .066$, $P = .730$), which suggests that the memory for the episodic binding between an object and a word does not linearly depend on recognition memory for the individual object.” (p 7/8)

Moreover “...the old/new recognition effects for the individual objects were not related to spindle activity ($r = .103 - .331$, $P = .074 - .589$), which provides evidence for the view that recognition memory for a specific object and episodic memory for the pairing of a specific objects with a word are supported by distinct consolidation mechanisms during infant sleep.” (p9)

In addition, we analyzed the object repetition effect in the encoding session and included the following paragraph at pages 8/9:

“In a next step, we explored, how the strength of encoding affected infant memory. The repetition of individual objects during the encoding session was reflected in the reduction of the negativity in the 600 – 800 ms time window of the ERP (Repetition $F_{1,58} = 12.229$, $P = .001$, $\eta_p^2 = .174$; Fig. S2a), an effect that was present in both groups (wake: $t_{29} = 2.302$, $P = .029$, $d = .420$, nap: $t_{29} = 3.119$, $P = .004$, $d = .569$ for the frontal-central maximum; Fig. S2d) and did not differ between them ($t_{58} = .063$, $P = .950$). This object encoding effect was not correlated with the frontal-central old/new object memory effect of the nap group, which had the same latency, but an inversed polarity ($r = -.239$, $P = .204$). The corresponding old/new ERP difference in the memory test of the wake group, however, was correlated with the encoding effect of this group ($r = .423$, $P = .020$; Fig. S2c). This pattern suggests that the immediate memory for individual objects during the encoding session, was to some extent retained in the infants of the wake group, but underwent strong modifications during the retention phase of the nap group.”

The new results are shown in the supplemental Figure S2.

We very much thank the reviewer for this comments! We realize that the additional analysis of the picture processing reveals important additional insights, which nicely fit in the overall pattern and strengthens the manuscript.

(5) There are a couple instances where the authors present their post-hoc interpretation of the results almost as fact (e.g., on page 10 in the Discussion: “The answer is that newly formed episodic memories...”). I suggest some of this language be reworded to be more speculative in nature.

Authors’ response:

We carefully revised the manuscript and reworded some phrases. The above mentioned phrase, in particular, has been changed into: “The most probable explanation is...” (p13).

(6) Minor points:

a. Page 3, Introduction: “A still unsolved question is, on which factors...” - I find this sentence to be unclear and slightly confusing.

b. In Figures 2 and 3, I suggest listing all the electrodes included in PO and LFT to the figure caption.

c. Effect sizes should be reported.

Authors’ responses:

a. The sentence has been replaced.

b. We have listed the electrodes in the Figure caption.

c. Now, we report effect sizes for all significant effects.

We appreciate the reviewers' constructive comment and feel that the manuscript has improved. Many thanks!

Reviewers' Comments:

Reviewer #1:

Remarks to the Author:

The revised manuscript has been greatly-improved, demonstrating transparency and rigour. I have one more comment and some minor suggestions:

(1) The following is my understanding of the nap-group findings: New object-label mismatch was detected by the access of semantic memory, i.e. based on their semantic knowledge of the object. Old object-label mismatch was detected by the access of episodic memory, i.e. based on their pre-nap (episodic) experience of the object-label pairing. The lack of N400 effect suggests that the nap group did not access their semantic memory within the expected response time frame, which the authors suggested is due to recent episodic memory inhibiting semantic memory.

(a) In the discussion, it would be valuable to briefly discuss other possible interpretations, and then address the reasons why the inhibition account is the most plausible (p.13)

(b) The authors have discussed why a selective inhibitory mechanism might be useful for fine differentiation, but it remains unclear what kind of inhibitory mechanisms might be in place to support this. I understand that further research has to be done, but it would be useful to highlight some existing findings and models on the inhibition of semantic representations in adults and children (it could be designs including and/or excluding episodic memory formation) to support your interpretation the findings suggest inhibition of semantic processing as the most plausible account.

Minor suggestions to improve readability of the manuscript.

In the results section, section titles were confusing/sometimes overreaching. Here are some suggestions which I hope the authors find useful:

(1) The titles 'Semantic memories are available in infants of both groups' (line 122) and 'Semantic access differs between infants of the wake and nap groups' (line 135) are confusing because the N400 semantic context effects (in different conditions) were used to infer the availability as well as access of semantic memories in Line 122 and Line 135. Are 'availability' and access of semantic memories referring to the same thing? If not, please clarify. If yes, I suggest refining the section titles along the lines of:

(a) 'Semantic access for new objects observed in both wake and nap groups' (for line 122)

(b) 'Semantic access for old objects differ between wake and nap groups' (for line 135)

(2) 'New episodic memories were formed in the infants of the nap group' (line 152)---
overreaching interpretation. Suggestion: 'FTMR suggests formation of new episodic memories in the nap group'

(3) 'Frontal sleep spindles enhance episodic memories' (line 186) is overreaching. Suggestion:
'Frontal sleep spindles activity is correlated with episodic memories'

Reviewer #2:

Remarks to the Author:

I reviewed the previous version of this manuscript and I continue to think that this work has significant strengths that include addressing an important question, utilizing an interesting experimental paradigm, and providing novel and informative results. I appreciate the authors' responses to the constructive critiques raised across all reviews. However, there are two issues that I don't think were sufficiently dealt with, but they could be easily addressed in an additional revision.

1. I mentioned that the interpretation of the results was a bit overreaching at times. I think the current version of the manuscript still suffers, in part, from the same problem. I don't think there is direct evidence for an inhibitory process leading to the absence of a N400 in the Nap group in response to the very same words presented with old objects even in conjunction with a presence of an N400 in response to novel objects in the same group of infants. The reference to an inhibitory process is still present in the manuscript. For example, the Abstract still states that there is evidence of selective inhibition and semantic suppression, both of which cannot be conclusively established (see also lines 306-310). I think rewording these parts of the manuscript would be appropriate.

2. I respectfully disagree with the authors' response that the examination of behavioral manifestations of memory such as novelty preferences would not be informative or would amount to addressing a completely different question. The authors are correct that novelty preferences provide an indirect measure of memory in that they reflect an attentional bias toward novel stimuli and that other measures may have other limitations. However, these measures have been used in hundreds of informative studies of infants' memory. Drawing connections between the ERP data presented here and possible behavioral implications of these findings is important. Recognizing the absence of behavioral measures in this study as a limitation and elaborating on what we might observe behaviorally would not detract from the merits of the manuscript. Instead, these changes would help situate the present work more effectively within the broader literature.

Reviewer #3:

Remarks to the Author:

Most of my concerns have been addressed. I have a few further minor points:

I think the last two sentences of the abstract are still stated too strongly – I think the authors should make it more apparent that this is their hypothesized explanation for their findings.

Introduction: The first sentence of paragraph 3 (line 56) doesn't seem that relevant to the rest of the paragraph. The rest of the paragraph focuses on the idea that consolidation requires replay, which happens during sleep via spindles. To me, the first sentence implies that there are differences in how new information is consolidated that determines how it contributes to episodic or semantic memory, but that isn't further explained or elaborated upon.

There are grammatical errors throughout the manuscript. I recommend the text is thoroughly proofread before publication.

Results (line 114): Why is the semantic congruity x hemisphere p-value reported as $P < .007$ when the others are reported precisely (e.g., $p = .008$ and $p = .0002$)?

Results (line 170): The early-latency old/new occipital memory effect is Fig S2b lower (not upper) panel, correct? And then on line 190, this should be Fig S2a upper panel?

In the figure caption for Fig S2b, it may be worth noting that overall it looks like there is a missing late frontal-central old/new object recognition effect, but that is because the effect was only present in the nap group and with reversed polarity.

Methods (line 522): fuzziness should be fussiness

Methods (line 672): are the electrodes included in the central ROI mislabeled? Should it be C3, CZ, and C4?

Reviewer #1

The revised manuscript has been greatly-improved, demonstrating transparency and rigour. I have one more comment and some minor suggestions:

(1) The following is my understanding of the nap-group findings: New object-label mismatch was detected by the access of semantic memory, i.e. based on their semantic knowledge of the object. Old object-label mismatch was detected by the access of episodic memory, i.e. based on their pre-nap (episodic) experience of the object-label pairing. The lack of N400 effect suggests that the nap group did not access their semantic memory within the expected response time frame, which the authors suggested is due to recent episodic memory inhibiting semantic memory.

(a) In the discussion, it would be valuable to briefly discuss other possible interpretations, and then address the reasons why the inhibition account is the most plausible (p.13)

(b) The authors have discussed why a selective inhibitory mechanism might be useful for fine differentiation, but it remains unclear what kind of inhibitory mechanisms might be in place to support this. I understand that further research has to be done, but it would be useful to highlight some existing findings and models on the inhibition of semantic representations in adults and children (it could be designs including and/or excluding episodic memory formation) to support your interpretation the findings suggest inhibition of semantic processing as the most plausible account.

Authors' response: In response to the reviewer's comment, we have included a detailed discussion of the missing N400 effect. In particular, we have referred to the conditions for the absence of semantic priming in adults and have discussed possible mechanisms that might cause the absence of the N400 in the infants of the nap group. We now discuss three potential interpretations of the absence of the semantic priming effect. First, it may be passively omitted due to an early, attention-capturing effect of episodic memory. Alternatively, semantic priming may be impaired due to a modification of the semantic processing of the old objects. Or lastly, semantic processing may be actively inhibited as a result of a competitive process between episodic and semantic memories. The following is how the new section reads.

“Missing semantic context priming, despite appropriate lexical-semantic memories, may result from insufficient semantic processing of either the primes or the targets. In adults, semantic processing of words occurs passively through an automatic spread of activation, and the N400 word priming effect is eliminated only when the primes are not attended³². Accordingly, the objects, serving as the primes in the present study, may be responsible for the missing N400 effect. In the infants who had episodic memories available, the recognition of an old object might have shifted attention towards the recent episode, thereby omitting semantic processing of the object. Rather than pure omission, the focus on the recognized details of an old object might have dissociated the object from the generalized semantic representation, such that the assignment to this representation failed and semantic priming of the related word was prevented. Even if this unsuccessful memory access induced the formation of a more specific semantic representation, the immediately formed representation would not yet have been linked to a word. Thus, semantic word priming would not have been possible. It is also conceivable that old objects accessed existing semantic memories and triggered semantic word priming, but that priming was ineffective, i.e., the words as the targets may be responsible for the missing effect. In this scenario, semantic processing of the words was initiated, but episodic memories were activated in parallel and competed with semantic memory activation. As a result, semantic processing of the words was inhibited, such that semantic priming had no effect on word processing. From the present data it is not clear whether new episodic memories had already captured attention before semantic priming was initiated, or whether they inhibited semantic

processing after semantic pre-activation. Irrespective of the underlying mechanisms, our findings strongly suggest that newly formed episodic memories in the nap group interfered with semantic processing and, at least temporarily, prevented access to existing semantic representations, when words were perceived in their recent episodic context.” (p13-14)

We thank the reviewer for pointing out this previously under-specified part of the discussion. We feel that the manuscript has improved with the more detailed discussion.

Minor suggestions to improve readability of the manuscript.

In the results section, section titles were confusing/sometimes overreaching. Here are some suggestions which I hope the authors find useful:

(1) The titles ‘Semantic memories are available in infants of both groups’ (line 122) and ‘Semantic access differs between infants of the wake and nap groups’ (line 135) are confusing because the N400 semantic context effects (in different conditions) were used to infer the availability as well as access of semantic memories in Line 122 and Line 135. Are ‘availability’ and access of semantic memories referring to the same thing? If not, please clarify. If yes, I suggest refining the section titles along the lines of:

(a) ‘Semantic access for new objects observed in both wake and nap groups’ (for line 122)

(b) ‘Semantic access for old objects differ between wake and nap groups’ (for line 135)

Authors’ response: We thank the reviewer for her/his suggestions. Our original section titles were intended to primarily convey two things. First, to highlight the presence and accessibility of semantic memories in infants of both groups (indicated by any N400 effect, here specifically to pairings with new objects). Secondly, they were intended to highlight the missing current access of these semantic memories (despite their accessibility) in the infants of the nap group. Taking into account the 60 character subheading limit, we have now changed the section titles to:

(a) Semantic memories were present in infants of both groups

(b) Semantic access of old events differed between groups

Also, we have replaced the term “availability” in this section: “The occurrence of this N400 context effect clearly attests the existence of appropriate lexical-semantic memories of the stimulus material and the similar accessibility of these memories in the infants of the wake and nap groups.” (p6)

(2) ‘New episodic memories were formed in the infants of the nap group’ (line 152)--- overreaching interpretation. Suggestion: ‘FTMR suggests formation of new episodic memories in the nap group’

Authors’ response: In our opinion, this is not overreaching. The FTMR of the nap group represents an ERP response that was specifically elicited by the reoccurrence of a word in the identical object context as during encoding. Such a response would require the presence of a detailed memory of the recent episodic co-occurrence of the object and the word. To emphasize this specificity, we added on page 7: “This fronto-temporal memory response (FTMR) did not occur for pairings with new objects ($t_{29} = -1.061$, $P = .298$).”

(3) ‘Frontal sleep spindles enhance episodic memories’ (line 186) is overreaching. Suggestion: ‘Frontal sleep spindles activity is correlated with episodic memories’

Authors’ response: We agree that this was overreaching. Following the limitation for subheadings, we have changed this title to: “Frontal sleep spindles are linked to episodic memories” (p10).

Reviewer #2

I reviewed the previous version of this manuscript and I continue to think that this work has significant strengths that include addressing an important question, utilizing an interesting experimental paradigm, and providing novel and informative results. I appreciate the authors' responses to the constructive critiques raised across all reviews. However, there are two issues that I don't think were sufficiently dealt with, but they could be easily addressed in an additional revision.

(1) I mentioned that the interpretation of the results was a bit overreaching at times. I think the current version of the manuscript still suffers, in part, from the same problem. I don't think there is direct evidence for an inhibitory process leading to the absence of a N400 in the Nap group in response to the very same words presented with old objects even in conjunction with a presence of an N400 in response to novel objects in the same group of infants. The reference to an inhibitory process is still present in the manuscript. For example, the Abstract still states that there is evidence of selective inhibition and semantic suppression, both of which cannot be conclusively established (see also lines 306-310). I think rewording these parts of the manuscript would be appropriate.

Authors' response: A related comment has been made by Reviewer 1 (comment #1) who asked for more elaborate discussion of alternative explanations for the missing N400 effect when words were presented with old objects. In response to the reviewer's comment, we have now carefully revised the manuscript in order to avoid overreaching. In particular, we have avoided the term "inhibition" throughout the text and also replaced the term "suppression". The abstract and the relevant parts of the discussion have been reworded as follows:

"We propose that temporarily disabled semantic processing protects precise episodic memories from interference with generalized semantic memories. Selectively restricted semantic access could also trigger semantic refinement, and thus, might even improve semantic memory." (Abstract)

"At first glance, restricted access to semantic memory does not seem to be favourable. In a changing environment, however, the most recent experience may be particularly relevant for behavioural adaptation in the near future. Thus, fast direct access to recent information without involvement of older memories might be advantageous. The unavailability of semantic memories may enable infants to re-experience recent events without any distortion by existing general knowledge and, thus, may protect precise episodic memories from interference with generalized semantic memories. Moreover, during earliest developmental stages, semantic long-term memory comprises extremely broad categories that act as strong attractors for a variety of objects and events and causes the well-known overgeneralizations in the comprehension and production of early words^{48,49}. The temporary selective restriction of access to existing representations might provide an opportunity to overcome strong attractors by enabling the formation of more specific representations. This potential mechanism would ultimately boost the development of infant semantic memory." (p14)

Also, as mentioned above, we have discussed alternative interpretations for the missing N400. See the response to Reviewer 1 (comment #1).

(2) I respectfully disagree with the authors' response that the examination of behavioral manifestations of memory such as novelty preferences would not be informative or would amount to addressing a completely different question. The authors are correct that novelty preferences provide an indirect measure of memory in that they reflect an attentional bias toward novel stimuli and that other measures may have other limitations. However, these measures have been used in hundreds of

informative studies of infants' memory. Drawing connections between the ERP data presented here and possible behavioral implications of these findings is important. Recognizing the absence of behavioral measures in this study as a limitation and elaborating on what we might observe behaviorally would not detract from the merits of the manuscript. Instead, these changes would help situate the present work more effectively within the broader literature.

Authors' response: We agree with the reviewer that in general, behavioural implications help to situate neurophysiological findings more effectively within developmental research. However, in our specific experimental design, the presence of newly formed episodic memories is expected to manifest itself in the same behavioural recognition effect as the presence of pre-existing or newly formed semantic memories. Meaning, the behavioural correlates of our effects would be identical. In our reading of the situation, direct behavioural implications are not possible for the present study, at least how it is currently designed. Perhaps a future study could devise a way to acquire a behavioural index that would vary across conditions. Nevertheless, we can try to tie our results in with more behavioural findings from previous work. Our results appear to be closely related to experimental findings of young children's mutually exclusive selection behaviour in the context of learning hierarchically related concepts. We have included the following paragraph into the discussion:

*“Selective exclusion of assignment to existing memories can in fact be observed in the context of early lexical-semantic specification. When young children have learned a new word as a label for a new sub-category of a known basic-level category, many of them subsequently avoid selecting objects of the sub-category as an instance of the basic-level category. For example, after learning the word *zav* as a name for tulips, about half of the children no longer identified tulips as flowers⁵⁰. This strong behavioural tendency to treat hierarchically related categories as mutually exclusive sub-categories, observed in somewhat older children, bears some similarity with the apparent mutually exclusive access to episodic and semantic memories observed in the present study. However, the precise relationships between the underlying memory processes and respective behavioural and ERP indicators still need to be explored by future research.” (p14/15)*

We're grateful to the reviewer for this contribution and feel that our main novel finding is now better situated in developmental research.

Reviewer #3

Most of my concerns have been addressed. I have a few further minor points:

(1) I think the last two sentences of the abstract are still stated too strongly – I think the authors should make it more apparent that this is their hypothesized explanation for their findings.

Authors' response: We have rephrased the last two sentences of the abstract: “We propose that temporarily disabled semantic processing protects precise episodic memories from interference with generalized semantic memories. Selectively restricted semantic access could also trigger semantic refinement, and thus, might even improve semantic memory.”

(2) Introduction: The first sentence of paragraph 3 (line 56) doesn't seem that relevant to the rest of the paragraph. The rest of the paragraph focuses on the idea that consolidation requires replay, which happens during sleep via spindles. To me, the first sentence implies that there are differences in how new information is consolidated that determines how it contributes to episodic or semantic memory, but that isn't further explained or elaborated upon.

Authors' response: We have reworded this sentence as a general statement to motivate our choice to focus on (sleep-dependent) consolidation: "The contribution of an experience to a certain kind of memory depends on both the encoding of relevant information and the consolidation of immediately formed memories." (p3)

(3) There are grammatical errors throughout the manuscript. I recommend the text is thoroughly proofread before publication.

Authors' response: The manuscript has been proofread by a native speaker.

(4) Results (line 114): Why is the semantic congruity x hemisphere p-value reported as $P < .007$ when the others are reported precisely (e.g., $p=.008$ and $p=.0002$)?

Authors' response: This was a mistake, we have corrected it to $P = .007$.

(5) Results (line 170): The early-latency old/new occipital memory effect is Fig S2b lower (not upper) panel, correct?

Authors' response: This is correct. Nevertheless, we no longer refer to individual panels.

(6) And then on line 190, this should be Fig S2a upper panel?

Authors' response: The main effect relates to both upper and lower panel of Fig. S2a.

(7) In the figure caption for Fig S2b, it may be worth noting that overall it looks like there is a missing late frontal-central old/new object recognition effect, but that is because the effect was only present in the nap group and with reversed polarity.

Authors' response: We have rephrased the figure caption for the Supplementary Figure 2b: "ERP responses of the overall group to old (black lines) and new (grey lines) objects at the memory test. Early occipital old/new object recognition effect (150 – 300 ms) indicating visual object memory. Missing late object recognition effect in the overall group due to the polarity-inversed ERP responses in the wake and nap groups."

(8) Methods (line 522): fuzziness should be fussiness

Authors' response: Changed.

(9) Methods (line 672): are the electrodes included in the central ROI mislabeled? Should it be C3, CZ, and C4?

Authors' response: Yes, they were. Thank you very much! We have corrected the labels.

We thank the reviewer very much for her/his very helpful and detailed notes.

Reviewers' Comments:

Reviewer #1:

Remarks to the Author:

The authors have satisfactorily responded to my one major comment and minor suggestions. Specifically, they included a detailed discussion of the potential interpretations of the missing N400 effects. They have also improved the overall readability of the manuscript. I would recommend acceptance of this report, which I believe is a valuable contribution to this area of research.

Reviewer #2:

Remarks to the Author:

I am satisfied with the revisions, which tempered the interpretation of the findings

Reviewer #3:

Remarks to the Author:

My concerns have been addressed and the manuscript has been greatly improved. I have no further comments.